# The Association between Cyberbullying Victimization and Depression among Children: A Moderated Mediation Model

**DOI:** 10.3390/bs14050414

**Published:** 2024-05-15

**Authors:** Kuai Song, Feng-Juan Zhou, Geng-Feng Niu, Cui-Ying Fan, Zong-Kui Zhou

**Affiliations:** 1Key Laboratory of Adolescent Cyberpsychology and Behavior (CCNU), Ministry of Education, Wuhan 430079, China; songpsy@mails.ccnu.edu.cn (K.S.); zhoufj@mails.ccnu.edu.cn (F.-J.Z.); niugfpsy@mail.ccnu.edu.cn (G.-F.N.); fancy@mail.ccnu.edu.cn (C.-Y.F.); 2School of Psychology, Central China Normal University, Wuhan 430079, China; 3Department of Preschool Education, Hubei Preschool Teachers College, Ezhou 436032, China; 4Administrative Office, Wuhan University, Wuhan 430072, China

**Keywords:** cyberbullying victimization, self-perceived social competence, optimism, depression, children

## Abstract

Cyberbullying victimization is becoming more prevalent and adversely affects mental health. This research explores the relationship between the two variables and the underlying mechanism, especially for children, as the impact of mental health in childhood might last a lifetime. Primary school students (*N* = 344; *M*_age_ = 9.90; 43.90% girls) completed self-report questionnaires regarding cyberbullying victimization, self-perceived social competence, optimism, and depression at school. Gender and grade were controlled as covariates. Depression was positively predicted by cyberbullying victimization, while self-perceived social competence played a partially mediating role. In addition, optimism directly and indirectly moderated the effects of cyberbullying victimization on depression. Specifically, the effects were stronger for children with low levels of optimism. Therefore, efforts to enhance children’s self-perceived social competence and optimism may reduce their depression resulting from cyberbullying victimization.

## 1. Introduction

Depression is one of the most common mental disorders around the world, with a predominant increasing trend in its prevalence over time [1]. The same is true of the increasing trend of depression in children. The overall incidence of depression increased substantially in both boys and girls between 2003 and 2018 [2]. Children are more likely to experience high rates of depression during and after the COVID-19 pandemic [3]. According to a meta-analysis, depression has become a common mental health problem in children [4]. The prevalence of depressive symptoms among children in China was 17.5% [5]. A long-term state of emotional maladjustment such as depression will pose a threat to children’s school performance, interpersonal communication, academic performance, and physical and mental health and may lead to suicidal ideation or even suicidal behavior in serious cases [6,7]. Depression in childhood may trigger developmental and adjustment problems in adolescence and adulthood [8,9]. Exploring the mechanism and influencing factors of childhood depression and carrying out timely interventions are vital for the physical and mental development of children.

The Internet has become an integral part of the ecosystem that influences the development and adaptation of individuals. In 2022, the Internet penetration rate of Chinese primary school students was 95.1%, with 38.7% of minors spending more than half an hour online on weekdays and 72.3% on holidays [10]. However, the Internet is a double-edged sword due to negative phenomena such as cyberbullying [11]. According to a comprehensive review, the prevalence of global cyberbullying among adolescents and children increased from 2015 to 2019, and the average global cyberbullying victimization rate was 33.08%, ranging from 13.99 to 57.5% [12]. The prevalence of cyberbullying victimization varied greatly among Chinese adolescents and children. A study of urban secondary school students in Guangdong province, China, reported a cyberbullying victimization rate of 44.5% [13], while the rate in rural primary and secondary school students in Shandong province, China, was 11.4% [14]. Variations in research tools and instruments, demographic characteristics of the survey sample, economic development, geographical location, cultural backgrounds, and Internet penetration rates may contribute to different cyberbullying victimization rates [15]. Researchers found that cyberbullying can cause many psychological adaptation issues (e.g., anxiety and depression) [16]. Prior studies have concerned traditional bullying and cyberbullying in adolescents and adults [17,18,19,20,21], but research on cyberbullying victimization among children is limited. However, it is worrisome that children are vulnerable to cyberbullying victimization because they have less experience in dealing with negative online events (compared to adolescents and adults). Therefore, timely and effective prevention and intervention measures in childhood can help individuals better cope with the long-term adverse effects of cyberbullying. The purpose of this study was to explore the effects of cyberbullying victimization on depression and its mechanism in children and to provide empirical evidence for the prevention of the negative effects of cyberbullying and depression on children.

According to the General Strain Theory [22], stressful life events increase the likelihood that individuals will experience a range of negative emotions. Cyberbullying, as a negative life event, is a source of stress that causes individuals to produce negative emotions. The characteristics of cyberbullying include many potential audiences, beyond time and space, and the invisibility of perpetrators [23]. Due to these characteristics, cyberbullying victimization is considered a greater risk for psychological and behavioral adaptation than traditional bullying victimization [24]. Cyberbullying victimization can cause individuals to feel rejected and isolated [25], and their need for belonging is difficult to meet. At the same time, the characteristics of cyberbullying (e.g., anonymity, fast, and widespread) may cause individuals to feel that the external world is uncontrollable, leading to helplessness, an important predictor of depression [26]. Meta-analyses and longitudinal studies have shown that cyberbullying victimization is a predictor of depressive symptoms among adolescents [16,27,28]. A 12-month longitudinal study indicated that cyber victimization significantly predicted depressive symptoms in young adolescents [29]. Therefore, we hypothesize that depression is positively predicted by cyberbullying victimization in children (H1). The mechanism underlying cyberbullying victimization and depression is another research focus. Understanding the underlying mechanisms can provide a basis for intervening in the harmful effects of cyberbullying victimization. Therefore, this study aims to further explore how cyberbullying victimization affects depression and potential individual differences in children.

### 1.1. Self-Perceived Social Competence as a Mediator

Many theories of depression posit that low self-esteem is a vulnerability factor involved in the etiology of depressive disorders [30]. According to Harter’s self-esteem theory, children’s self-esteem comprises scholastic competence, social acceptance, athletic competence, physical appearance, and behavioral conduct [31]. Self-perceived social competence is an individual’s self-esteem in the social acceptance domain, which is generally defined as a belief in one’s ability to perform competently in a social situation [32]. Individuals with low self-perceived social competence may feel less competent in maintaining harmonious interpersonal relationships and dealing with interpersonal conflicts. Previous studies have found that low self-perceived social competence can significantly predict an increase in depression [33,34].

Cyberbullying is a negative event that occurs in interpersonal relationships via electronic means. Negative interpersonal experience may impair social self-perception, because building social self-perception relies on interpersonal interactions and social feedback [35]. Children’s self-evaluation is developmental and especially vulnerable to social context and interpersonal experiences [36]. Experiencing adverse interpersonal events (e.g., cyberbullying) may produce negative feelings of unacceptability and worthlessness [37]. These negative feelings may produce self-doubt, self-denial, and pessimistic beliefs about social competence. According to the competency-based model of depression, negative events foster the development of maladaptive self-cognitions. Children may construct a negative sense of self when they face negative self-information from negative events [38]. Children with poor perceptions of their competence are at an increased risk of developing depressive symptoms [39]. In addition, a longitudinal study also found that cyber victimization predicted negative self-cognition [29]. Therefore, cyberbullying victimization may detrimentally influence self-perceived social competence. Studies showed that children’s social self-perceptions mediated the effect of peer difficulties on internalizing problems [40,41]. Thus, it might be assumed that cyberbullying victimization predicts depression through the mediating role of self-perceived social competence (H2).

### 1.2. Personality Trait of Optimism as a Moderator

According to the diathesis–stress theory of depression, the interaction between the environment and the individual determines mental health [42]. Positive personality traits (e.g., resilience, growth mindset, and mental capital) are protective against stressful environments and can reduce the occurrence of psychological problems [43,44,45]. Optimism as a positive personality trait is a protective factor for an individual’s mental health. Thus, it is important to explore optimism’s role in the relationship between cyberbullying and depression.

According to the cognitive vulnerability model of depression, people’s cognitive styles and the way they typically interpret or explain events in their lives significantly affect their vulnerability to depression; negative cognitive styles are present in all types of depression [46]. Optimism is a generalized tendency toward positive expectations of the future and life engagement [45,47]. Optimists are biased toward positive information and can look at the positive aspects in the face of stress [48]. Additionally, optimists tend to employ positive cognitive strategies (e.g., cognitive reappraisal) and coping strategies (e.g., problem-solving) to mitigate the effects of adversity on negative psychological outcomes [49,50,51]. Individuals with an optimistic attitude tend to interpret negative situations as external, temporary, and specific reasons [52]. This interpretation approach enables individuals to have a more comprehensive view of the problem, reducing the risk of emotional and behavioral problems. When faced with cyberbullying victimization, optimists accept their situation realistically and adaptively solve problems. Furthermore, optimists also have greater self-confidence and perseverance against cyberbullying victimization [53]. That is, optimism can reduce vulnerability to depression by increasing positive thoughts, adaptive coping, and perseverance. In addition, cross-sectional and longitudinal studies have found that optimism moderated the relation between stress (e.g., perceived psychological stress, job stress, and coronavirus stress) and depression [54,55,56]. Thus, one might assume that optimism moderates the direct and indirect effects of cyberbullying victimization on depression. Compared with high optimism, cyberbullying victimization could, directly and indirectly, predict depression through self-perceived social competence in low optimism (H3).

Based on the stress process model [57], the competency-based model of depression [38], and the diathesis–stress theory [42], the undesirable impacts of stressors (e.g., cyberbullying victimization) can be mediated by self-esteem factors (e.g., self-perceived social competence) and moderated by personality traits (e.g., optimism). As a result, we developed a moderated mediating model (Figure 1) based on relevant studies to explain the relationship between cyberbullying victimization and depression among children—considering self-perceived social competence as a mediator and optimism as a moderator.

## 2. Method

### 2.1. Participants and Procedure

Using a convenience sampling method, we collected data from the primary school attached to Central China Normal University in Hubei province, which is one of China’s top 100 primary schools. The students in this primary school are the children of Central China Normal University staff and residents of the nearby community. We reached out to the vice principals of the primary school. As young children may have a limited cognitive ability to read and understand the questionnaire content, we covered all grade three to six students at this primary school. We recruited eight pre-trained psychology graduate students as experimenters. The training included learning the questionnaire content, responses to inquiries, interpersonal communication, and coding the questionnaire. Then, the experimenters collected the paper–pencil questionnaires at the primary school in April 2021. This study involving human participants was reviewed and approved by the Ethical Committee of Central China Normal University for Compliance with Ethical Standards (CCNU-IRB-201611025b).

Participants were informed of the confidentiality of the data. We also explained informed consent and the right to withdraw at any time to the children. The children were asked to respond independently and authentically, and they completed a set of questionnaires in the classroom. Throughout the process, the participants did not show difficulty in understanding the content of the questionnaire. Finally, this study comprised 344 participants (*M_age_* = 9.90, *SD* = 1.20; 43.90% girls), and their details are given in Table 1.

### 2.2. Measures

#### 2.2.1. Cyberbullying Victimization

In this study, we employed the Chinese version of the Cyberbullying Scale [58], which was adapted from the general cyberbullying measures by Hinduja and Patchin (2008) [59]. This scale has high reliability and validity in Chinese participants [58,60]. It is divided into two dimensions with 8 items in total, namely, cyberbullying and cyberbullying victimization. The present study adopted 4 items from the cyberbullying victimization dimension (e.g., “Have you ever been bullied by others online or through text messages?”). The subscale adopted a four-point Likert scale, with higher scores indicating greater cyberbullying victimization. Scores for cyberbullying victimization were averaged across items. The subscale’s Cronbach’s alpha was 0.85 in the current study.

#### 2.2.2. Self-Perceived Social Competence

We applied the Perceived Competence Scale for Children [61]. The scale was adopted and validated for Chinese children [62] and has 6 items. The participants were presented with two descriptive sentences simultaneously on the right and left sides to offset the tendency toward socially desirable responses. An example of an item is “Some children find it difficult to make friends” vs. “Other children find it easy to make friends”. First, participants chose which side of the description fit them better. Second, they reported the degree of truth of the side of the description for them. This scale adopted a four-point Likert scale. The higher the score, the higher the self-perceived social competence. Items were averaged to form a subscale score. The questionnaire’s Cronbach’s alpha was 0.63 in the current study.

#### 2.2.3. Optimism

We adopted the Chinese version [63] of the Life Orientation Test-Revised [64]. This scale has been used in previous studies as a measure of dispositional optimism and has good reliability and validity [44,65]. The Chinese version has 12 items and adopts a five-point Likert scale, with “1” indicating strongly disagree and “5” indicating strongly agree. It contains five positively phrased items (e.g., “I always thought I was going to be lucky”), five negatively phrased items (e.g., “I think things rarely follow my ideas”), and two filler items (e.g., “Relaxation is easy for me”). Item responses were averaged to form the optimism score, with higher scores indicating greater optimism. The questionnaire’s Cronbach’s alpha was 0.81 in the current study.

#### 2.2.4. Depression

Many studies have adopted the Center for Epidemiologic Studies Depression Scale (CES-D) [66]. We applied the Chinese version [67], which has high reliability and validity in Chinese participants [68]. It contains 20 items and uses a four-point Likert scale. Item responses were averaged to form the depression score, with higher scores indicating higher levels of depression. The questionnaire’s Cronbach’s alpha was 0.91 in the current study.

### 2.3. Statistical Analysis

We adopted SPSS 20.0 to analyze the data collected in this study. First, all the variables were analyzed by Pearson’s correlation analyses to examine the potential relationships among variables, including cyberbullying victimization, self-perceived social competence, optimism, and depression. Second, to investigate the status of cyber victimization and depression in Chinese children, we used MANOVA to explore children’s gender and grade differences in all variables. The PROCESS macro for SPSS (Model 15) [69] was applied to analyze the proposed moderated mediation model.

## 3. Results

### 3.1. Descriptive Statistics and Correlations

In total, 31.1% (*n* = 107) of the children reported experiencing cyberbullying victimization at least once, including 70 boys (36.3%) and 37 girls (24.5%), 20 participants from grade 3 (27.8%), 23 participants from grade 4 (25.0%), 24 participants from grade 5 (27.0%), and 40 participants from grade 6 (44.0%). We used MANOVA to examine the main effects and interaction effects of gender and grade. Results of MANOVA for gender and grade are presented in Table 2. The results showed a significant main effect of grade (*F* _(3,336)_ = 4.70, *p* = 0.003, ƞ_p_^2^ = 0.04) on cyberbullying victimization. Post hoc tests suggested that the sixth graders experienced greater cyberbullying victimization than third graders (*p* = 0.006), fourth graders (*p* = 0.001), and fifth graders (*p* = 0.019). Additionally, the analysis also showed a significant interaction effect of gender and grade on depression among children (*F* _(3,336)_ = 3.06, *p* = 0.028, ƞ_p_^2^ = 0.03). Then, simple effect analysis indicated that boys in lower grades reported more serious depression than girls, and the opposite was true for upper grades.

The results of descriptive statistics and the correlation matrix of all variables are presented in Table 3. Cyberbullying victimization was positively related to depression. Cyberbullying victimization was negatively related to self-perceived social competence and optimism. Self-perceived social competence was positively related to optimism and negatively related to depression. Additionally, optimism was negatively related to depression.

### 3.2. Testing for the Moderated Mediation Model

As presented in Table 4, the regression models included gender and grade as covariates. Cyberbullying victimization negatively predicted self-perceived social competence (β = −0.11, *p* = 0.042) and positively predicted depression (β = 0.18, *p* < 0.001). Depression was negatively predicted by self-perceived social competence (β = −0.19, *p* < 0.001). These results suggested that cyberbullying victimization positively predicted depression in children, supporting H1. Additionally, self-perceived social competence partially mediated the effect of cyberbullying victimization on depression, supporting H2.

Moreover, depression was marginally significantly predicted by the interaction of cyberbullying victimization and optimism (β = 0.09, *p* = 0.050). This finding showed that optimism moderated the relationship between cyberbullying victimization and depression. Depression was significantly predicted by the interaction of self-perceived social competence and optimism (β = 0.12, *p* = 0.005). This finding showed that optimism moderated the relationship between self-perceived social competence and depression. Then, we further adopted simple slope analysis. As presented in Figure 2 and Figure 3, there was a significant effect of cyberbullying victimization (β_simple_ = 0.26, *p* < 0.001) and self-perceived social competence (β_simple_ = −0.29, *p* < 0.001) on depression in children with optimism at 1 *SD* below the mean. However, the effect of cyberbullying victimization (β_simple_ = 0.10, *p* = 0.134) and self-perceived social competence (β_simple_ = −0.09, *p* = 0.181) was non-significant when the optimism values were 1 *SD* above the mean.

The conditional indirect effects were examined at three values of optimism, including the mean and one standard deviation above and below the mean. The results showed that the indirect effects were significant and positive only at *M* − 1*SD* and *M* values of optimism. However, the indirect effect was not significant at *M* + 1*SD* values of optimism. Thus, the results supported H3.

## 4. Discussion

Our study found that the rate of cyberbullying victimization among Chinese children was 31.1%. There was no significant gender difference in cyberbullying victimization of children. Cyberbullying victimization increased with grade, and the sixth graders experienced significantly more cyberbullying victimization than other children. As the grade increases, children spend more time online and use more information and communication technology. At the same time, with the development of technological innovation and Internet applications, children are more likely to be involved in cyberbullying and become victims of cyberbullying. Previous studies of cyberbullying victimization have shown inconsistent findings regarding gender differences, which may be related to variations in the study features (e.g., modality of cyberbullying, cyberbullying scale type, representativeness of sampling, region of sampling, and reporting time frame) [70]. Our research found that boys in lower grades reported more serious depressive symptoms than girls, while girls in upper grades reported more serious depressive symptoms than boys. Boys’ social support networks are less intimate and supportive than girls’ [71]. Girls showed more positive emotions than boys in middle childhood [72]. Boys were more likely to interact with information and communication technology, increasing their risk of cyberbullying victimization. These may explain why depressive symptoms are higher in lower-grade boys. Girls in upper grades reached puberty earlier than boys, which is an important factor affecting girls’ depression [73]. Moreover, girls in upper grades are more relationally oriented and exhibit greater affiliative needs [74]. As the grade increases, girls are more likely to use social media and become depressed due to cyberbullying victimization than boys.

This study explored the effect of cyberbullying victimization on depression and its underlying process in children by constructing a moderated mediation model. The results indicated that self-perceived social competence was a mediator and optimism was a moderator in the direct and indirect effects. Specifically, compared with high optimism, cyberbullying victimization could directly predict depression in low optimism. Cyberbullying victimization also could indirectly predict depression through self-perceived social competence in low optimism.

Our study found that cyberbullying victimization positively predicted depression in children, which is consistent with previous research on the relationship between negative life events and depression [19,21]. According to the General Strain Theory [22], as a repeated, long-term, and uncontrollable stressor, cyberbullying can arouse strong negative emotions in victimized children. The more cyberbullying they experience, the more depressive symptoms they will have. This finding extends previous research by focusing on children. Similar to findings in adults and adolescents, cyberbullying victimization may also trigger depression in children.

### 4.1. The Mediating Role of Self-Perceived Social Competence

We found that self-perceived social competence partially mediated the relationship between cyberbullying victimization and depression in children. This finding aligns with the competency-based model of depression [38]. Additionally, it generally agrees with previous research examining the relationships among negative events, self-system beliefs, and mental health [75,76]. The findings expand upon previous research by focusing on children and negative online events.

According to the competency-based model of depression, negative events foster the development of maladaptive self-cognitions. When children are confronted with high levels of negative information about themselves, they are more likely to construct a negative sense of self, including feelings of incompetence. These negative self-constructs may engender feelings of depression [39]. The discovered mediating effect indicates that cyberbullying victimization influences children’s self-system belief in social competence and in turn develops into internalizing symptoms (i.e., depression). Children have an immature function of evaluation. Cyberbullying can cause children to develop negative self-evaluation, the core symptom of depression. Children’s appraisal of social competence is developing and easily affected by peer victimization [77]. Children who are mocked by peers repeatedly will doubt their social ability [33]. Additionally, Chinese culture emphasizes collectivism and interpersonal harmony [78]. Self-perceived social competence is closely associated with an individual’s need for belonging and competence, which are essential to children. Children with poor social self-competence may feel unaccepted by others and worthless in the social domain, and they may develop depression. Therefore, cyberbullying victimization reduces children’s self-perceived social competence. Poor self-perceived social competence makes individuals vulnerable to depression. Given the above, interventions should be used to enhance self-perceived social competence to minimize the detrimental influences of cyberbullying on children.

### 4.2. The Moderating Role of the Personality Trait of Optimism

Consistent with our hypotheses, the results showed that optimism directly and indirectly moderated the effect of cyberbullying victimization on depression. Specifically, children with lower optimism were more prone to experiencing depression resulting from cyberbullying victimization. Additionally, at high levels of optimism, self-perceived social competence did not play a mediating role. However, at low levels of optimism, cyberbullying victimization could influence depression through the mediating role of self-perceived social competence. These results align with the diathesis–stress theory, the cognitive vulnerability model of depression, and studies about optimism [44,45,46]. Compared with previous research, this research further expanded on the protective effect of optimism by focusing on children and interpersonal stressors in the virtual environment (i.e., cyberbullying victimization).

We found that optimism can moderate the effect of cyberbullying victimization on depression. Children with different levels of optimism respond differently to cyberbullying victimization. Optimists maintain a positive tendency to expect good outcomes even in stressful situations. Compared with children with lower optimism, children with higher optimism focus on the positive features of stressful situations and cope with stressful situations effectively. This positive cognitive and coping style can act as a buffer between stressful situations and emotional problems in children. Namely, optimists respond positively to cyberbullying and therefore have a lower risk of depression.

Moreover, optimism could buffer the effect of low self-perceived social competence resulting from cyberbullying victimization on depression. Children may construct a negative sense of self when they face negative self-information from negative events [38]. Experiencing cyberbullying victimization can significantly predict decreased social self-perceived social competence. As a positive psychological resource, optimism is usually durable [79]. That is, it denotes stable and positive general outcomes that are largely independent of self- and relationship-based processes [49]. There are two explanations for the moderating role of optimism on indirect pathways. First, optimists always regard future-oriented expectations with a positive attitude [47]. Positive expectations give children the confidence to boost low social competence. According to expectancy–value models of motivation, the value of outcomes (e.g., self-perceived social competence is important) and positive expectations (e.g., self-perceived social competence can be enhanced) can help individuals increase their motivation and strive to overcome difficulties [80]. For children with low self-perceived social competence, if they have high levels of optimism and believe that peer interaction and relationships can lead to good outcomes, they will work harder and more effectively at their relationships. This improves the individual’s self-perceived social competence and positive emotions, thereby reducing the risk of depression. Second, optimism, as a dispositional trait, predisposes individuals to use specific coping styles during peer interaction. Optimists are more likely to employ positive coping strategies to solve problems [45]. Optimism predicted more problem-focused coping (e.g., information seeking, active coping and planning, seeking benefits) with controllable stressors and more emotion-focused coping (e.g., acceptance, use of humor, positive reframing) with uncontrollable stressors [49]. Optimists may strive to learn social skills and seek social support to improve their self-perceived social competence and further prevent depression. However, fewer optimists tend to adopt avoidant coping strategies (e.g., suppression of thoughts, giving up, cognitive avoidance, self-distraction, focus on distress, and overt denial), which may exacerbate emotional symptoms. Thus, enhancing children’s optimism can reduce the adverse effects (e.g., depression) resulting from cyberbullying victimization.

### 4.3. Implications

This research is the first to explore the relationship between cyberbullying victimization and depression in children, as well as the underlying mechanisms (self-perceived social competence as a mediator and optimism as a moderator). The findings extended previous research by focusing on children and bullying victimization in virtual environments. The results are also practical for intervention programs for cyberbullied children. In addition, personality is developmental and shapeable during childhood [81]. Studies have indicated that self-perceived social competence and optimism can be trained and improved [82,83,84]. To reduce depression caused by cyberbullying, children (especially those who are cyberbullied) should be given methods to effectively raise their self-perceived social competence and optimism. Consequently, the effect of positive cognition on social competence and optimism could effectively prevent the detrimental impact of cyberbullying victimization. Centering on the important role of self-perceived social competence and optimism, this study is of theoretical value as it explores the harmful influence of cyberbullying victimization on children. At the same time, it is of great practical significance for prevention and intervention programs to alleviate depression in children.

### 4.4. Limitations and Future Directions

Firstly, we utilized a cross-sectional approach, so causal inferences must be made with caution. This study used a single source and a limited number of participants. We collected data from an urban primary school using a convenience sampling method, which limited the representativeness of the findings. Demographic characteristics of the survey sample, economic development, geographical location, cultural backgrounds, and Internet penetration rates may contribute to different cyberbullying victimization rates [15]. In the future, longitudinal studies with larger samples should be conducted to better explore whether and how childhood experiences of cyberbullying affect mental health later. Secondly, our survey methodology relied on self-report measures. The self-reported data can be subject to faulty and differential recall, intentional distortion, and inattention. The Cronbach’s alpha of the Perceived Competence Scale for Children in this study was 0.63, possibly due to having fewer items in the scale. The lower reliability of the scale may affect our results and their interpretation. Future studies may need to further validate the findings using other research methods, tools, and subject populations. Additionally, our study investigated cyberbullying victimization, without considering traditional bullying. Future research could explore differences in the underlying mechanisms between the two types of bullying. Finally, the finding of the partial mediating role of self-perceived social competence implies that other factors also play mediating roles. Additional mediators should be explored to fully reveal the potential mechanisms of the detrimental effects of cyberbullying victimization.

## 5. Conclusions

Overall, this study showed that depression was positively predicted by cyberbullying victimization. Self-perceived social competence was a partial mediator in the relationship between cyberbullying victimization and depression in children. The direct and indirect effects were moderated by optimism. Specifically, in children with low optimism, cyberbullying victimization not only directly predicted depression but also indirectly predicted depression through the mediating role of self-perceived social competence; whereas in children with high optimism, a relationship was not found.

## Figures and Tables

**Figure 1 behavsci-14-00414-f001:**
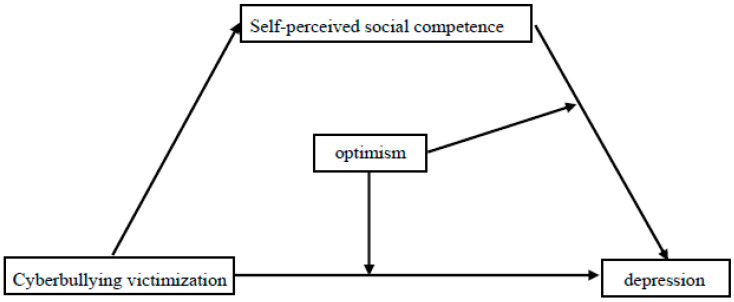
The proposed moderated mediation model.

**Figure 2 behavsci-14-00414-f002:**
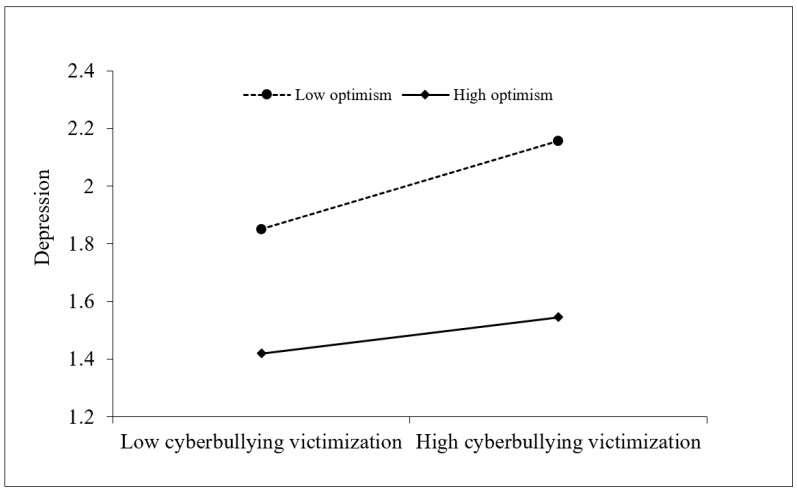
Optimism moderated the relationship between cyberbullying victimization and depression.

**Figure 3 behavsci-14-00414-f003:**
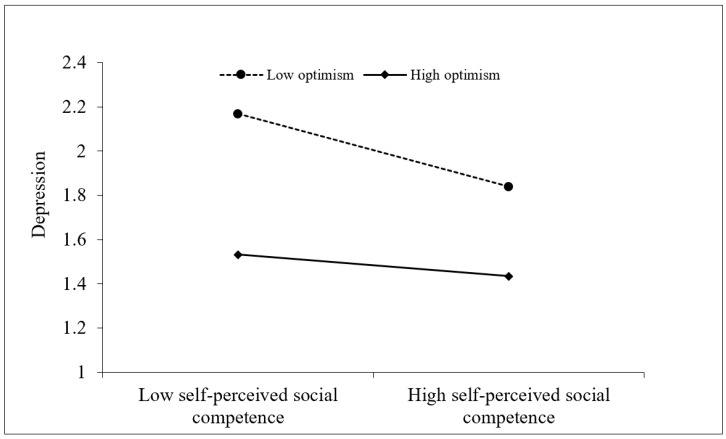
Optimism moderated the relationship between self-perceived social competence and depression.

**Table 1 behavsci-14-00414-t001:** Distribution of the number of boys and girls by grade.

	Grade Three	Grade Four	Grade Five	Grade Six	Total (N)
Boys (N)	35	52	48	58	193
Girls (N)	37	40	41	33	151
Total (N)	72	92	89	91	344

**Table 2 behavsci-14-00414-t002:** MANOVA of gender and grade.

Variables	*SS*	*df*	*MS*	*F*	*p*	ƞ_p_^2^
Outcome: CV						
Gender	1.32	1	1.32	3.88	0.050	0.01
Grade	4.79	3	1.60	4.70 **	0.003	0.04
Gender × Grade	1.04	3	0.35	1.02	0.384	0.009
Outcome: Depressive symptoms						
Gender	0.08	1	0.08	0.23	0.632	0.001
Grade	1.34	3	0.45	1.35	0.258	0.01
Gender × Grade	3.03	3	1.01	3.06 *	0.028	0.03

Note: * *p* < 0.05, ** *p* < 0.01. CV = cyberbullying victimization; *SS* = sum of squares; *df* = degrees of freedom; *MS* = mean square; ƞ_p_^2^ = effect size.

**Table 3 behavsci-14-00414-t003:** Descriptive statistics and intercorrelations between variables.

Variables	*M*	*SD*	1	2	3
1. Cyberbullying victimization	1.29	0.60			
2. Self-perceived social competence	2.88	0.57	–0.12 *		
3. Optimism	3.54	0.84	–0.21 ***	0.53 ***	
4. Depressive symptoms	1.64	0.58	0.33 ***	–0.47 ***	–0.61 ***

Note: * *p* < 0.05, *** *p* < 0.001.

**Table 4 behavsci-14-00414-t004:** Regressions testing self-perceived social competence as a mediator and optimism as a moderator in the relationship between cyberbullying victimization and depression.

Regression Models	β	*SE*	*t* Value	LLCI	ULCI	*R* ^2^	*F* Value
Outcome: SPSC						0.01	1.55
Predictors: Gender	0.01	0.06	0.16	−0.112	0.132		
Grade	−0.004	0.03	−0.15	−0.060	0.052		
CV	−0.11	0.05	−2.04 *	−0.209	−0.004		
Outcome: Depressive symptoms						0.48	43.44 ***
Predictors: Gender	0.04	0.05	0.79	−0.055	0.129		
Grade	−0.04	0.02	−1.99 *	−0.086	−0.001		
CV	0.18	0.04	4.09 ***	0.094	0.267		
SPSC	−0.19	0.05	−3.91 ***	−0.283	−0.094		
Optimism	−0.31	0.03	−9.38 ***	−0.377	−0.246		
CV × Optimism	−0.09	0.05	−1.97 ^+^	−0.181	0.000		
SPSC × Optimism	0.12	0.04	2.83 **	0.038	0.208		
Conditional indirect effect	Optimism values	Effect	Boot *SE*	BootLLCI	Boot ULCI		
	−1 (*M* − 1 *SD*)	0.03	0.02	0.002	0.067		
	0 (*M*)	0.02	0.01	0.002	0.044		
	1 (*M* + 1 *SD*)	0.01	0.01	−0.003	0.029		

Note: Gender was dummy coded (male = 1; female = 0). CV = cyberbullying victimization; SPSC = self-perceived social competence; LL = lower limit; CI = confidence interval; UL = upper limit. ^+^ *p =* 0.05, * *p* < 0.05, ** *p* < 0.01, *** *p* < 0.001.

## Data Availability

Data are available on request from the authors.

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
