# Peer review of "The Association between Cyberbullying Victimization and Depression among Children: A Moderated Mediation Model"

_behavsci, 2024, doi:10.3390/bs14050414_

Round 1

Reviewer 1 Report

Comments and Suggestions for Authors

Dear Authors,

I have read your paper entitled "The association between cyberbullying victimization and depression among children: A moderated mediation model." The topic of this manuscript is very current and unfortunately widespread among younger children. In terms of the qualities of this paper, I would particularly like to highlight the use of current literature and the focus on cyberbullying victimisation among children.

Despite numerous strengths, I must emphasise that the paper has several shortcomings and limitations that detract from its quality:

(1)     The manuscript requires proofreading and considerable revision to refine the English language. In its current form, it is quite difficult to follow the logic of the paper in some places due to unclear sentences.

(2)     In addition, the authors need to carefully read through and revise the entire paper, as some sentences in their current form are completely unclear to the reader and interfere with the fluidity and logical flow of the paper (examples of such sentences are: "Empirical studies have showed that cyberbullying victimization is positively predicted depression among adolescents." and "Participants were from 344 children from grade 3 to 6…").

(3)     It is necessary to revise the citation style in the text according to the instructions for authors of this journal.

Below are some of the key suggestions for each section of the paper.

INTRODUCTION

-            The second introductory sentence mentions "increasing pressure of competition" is mentioned ("With the rapid pace of social life and the increasing pressure of competition, children are facing multiple pressures and challenges, and there is an obvious younger-age trend of mental disorders (Meherali et al., 2021)."). I would like to ask the authors to explain more precisely what they mean by this.

-            I suggest that the authors reorganise the introductory part of the manuscript so that the existing scientific evidence is presented first. Then the main aim (or aims) of the research must be clearly emphasised and the research hypotheses presented at one point. Then the research gap that this study fills must be stated, or the scientific contribution. The current structure confuses the reader and the hypotheses are not clearly formulated.

METHOD

-            Cyberbullying victimization measure - in the section where the Cronbach's alpha is mentioned, do the authors mean the subscale (or dimension of victimization) and not the whole scale? If so, this needs to be clearly emphasised.

-            Perceived Competence Scale for Children - Cronbach's alpha is 0.63; this indicates a lower reliability of the scale. This also needs to be mentioned in the limitations of this study - the lower reliability of the scale may affect the results obtained and their interpretation.

RESULTS

-            This sentence is very unclear in its current version: "Second, the gender and grade differences were examined by MANOVA of conducting a 2 (Gender: male and female) × 4 (Grade: 3, 4, 5, and 6)." (lines 222-223)

-            I suggest presenting the results of the MANOVA analysis in tabular form so that the reader can follow the text more easily and check the results.

-            In Table 3, a "+" sign is inserted next to a result (specifically: - 1.97+). Is this an error or does it indicate something that is not clear what it refers to?

DISCUSSION

-            Revise the section on the limitations of the study (e.g. Cronbach alpha of the individual scales, convenience sample, etc.).

Based on all the above, in its current form, this manuscript is not suitable for publication in the Behavioral Sciences journal.

With respect,

Reviewer

Comments on the Quality of English Language

The manuscript requires proofreading and considerable revision to refine the English language. In its current form, it is quite difficult to follow the logic of the paper in some places due to unclear sentences.

Author Response

Dear reviewer,

Thank you very much for the suggestions, which made the manuscript better.

1.The manuscript has been given to MDPI's English editor for revision to make the sentences more fluent and logical.

2.I have rechecked the paragraphs and sentence expression of the manuscript, and have asked the editor with English background to revise and improve the whole manuscript. The following paragraphs have been revised (sentences are: "Empirical studies have showed that cyberbullying victimization is positively predicted depression among adolescents."  and "Participants were from 344 children from grade 3 to 6…").

  1. I have revised the references and citation style according to the requirements of the journal reference format.

4.In the introduction section, I expanded on the following sentence ("With the rapid pace of social life and the increasing pressure of competition, children are facing multiple pressures and challenges, and there is an obvious younger-age trend of mental disorders (Meherali et al., 2021).").  The aim is to point out the increasing risk of depression in children.

5.In the introduction section, I revised the content of the introduction to show the existing research results, and put forward the purpose and significance of this research on this basis, so as to make the content of this part clearer

6.In the method section,I have revised the expression of this part of the method, and made it clear that what we measure is the subscale of Cyberbullying victimization.

7.In the method section,In the revised manuscript, we mentioned the limitations of low reliability of Perceived Competence Scale for Children, which may be related to fewer items, and we hope that other research tool can be selected for future research.

8.In the results section, I put the following sentence "Second, the gender and grade differences were examined by MANOVA of conducting a 2 (Gender: male and female) × 4 (Grade:  3, 4, 5, and 6)." Change to "we used MANOVA to examine the main effects and interaction effects of gender and grade". This will convey the meaning more clearly

9.In the results section, thank you for your advice, I have presented the results of the MANOVA analysis in tabular form so that the reader can follow the text more easily and check the results.

10.In the results section, In Table 3, a "+" sign inserted next to a result (specifically: -1.97 +) is a symbol with marginal significance, and corresponding to these sentences “+p = 0.05.”, "depression was marginal significantly predicted by the interaction of cyberbullying victimization and. optimism (β = 0.09, p = 0.050)".

11.In the discussion section, I have revised the section on the limitations of the study (e.g. Cronbach alpha of the individual scales, convenience sample, etc.) and give some suggestions for future research.

Thank you again for your good advice to make this manuscript even better!

Reviewer 2 Report

Comments and Suggestions for Authors

Overall, the manuscript is well-written, and the methodology sounds robust. I have a few suggestions, as I documented below, to further strengthen argumentations and to streamline the text.

Title

Seeing that you used a convenience sample, I suggest to change the title a bit in order to clarify this: i.e.,

The association between cyberbullying victimization and depression in a group of Chinese children…

Introduction

Within the literature review, the author(s) might want to better justify their hypothesis by commenting on very recent publications on the current prevalence rates of cyberbullying and cybervictimization (also in China): i.e.,

Sorrentino, A., Sulla, F., Santamato, M., di Furia, M., Toto, G. A., & Monacis, L. (2023). Has the COVID-19 Pandemic Affected Cyberbullying and Cybervictimization Prevalence among Children and Adolescents? A Systematic Review. International Journal of Environmental Research and Public Health20(10), 5825.

Also, the results of this systematic investigation question the link between increased internet use and increased of cyberbullying/victimization (making what you write on lines 40-43 less harsh); and they provide the results of studies conducted in Chinese primary schools that report a high rate of children not involved in CB and/or CV.

Lines 73-74: “Empirical studies have showed that cyberbullying victimization is positively predicted depression among adolescents.” Please, revise the form of this sentence. This should either be ‘positively predicts’ o ‘is positively predicted by’, I guess.

Line 95: “Bullying is a negative event that occurs in interpersonal relationships.” You start writing about bullying, but then you actually report data on cyberbullying. Bullying and Cyberbullying are two different phenomena in several theorizations. I suggest to provide the definitions of traditional bullying and cyberbullying you refer to, as to clarify what you write on line 95.

Method: Participants and procedures

For the sake of replicability, I suggest to provide further information on the recruitment of the sample and data collection: e.g., how did you chose and contact the school? Was a convenience sample? This school can only be attended by CCN University staff’s children? In what period the data collection was done? Were the questionnaires completed online or in paper-pencil format? Have you provided a definition of cyberbullying to the children? What was it?

Was your investigation approved by any ethic committee? On this issue, how did you reduce the risk of inducing negative emotions in children after asking them about cybervictimization? Experimenter were graduate psychology students but not licenced psychologist. If a child would have need psychological support after being asked to remind whether he/she was cybervictimized, how did you cope with it?

Measures

I suggest to add references to the validation of the scales you used (i.e., cybervictimization; Self-perceived social competence) in your language; where not available, I suggest you to declare that you used translated but not validated measures and to address this also in the limitations section.

Discussions

I suggest to address in the discussions also the differences in cybervictimization and depression based on age and gender of your participants.

Once again, thank you for this investigation, as I find that the results of your studies would help future research in this field and that will make a difference in the increase of mental health in children. 

Comments on the Quality of English Language

English is fine. There is some typo here and there, that the authors should accurately check, but the quality of English is ok.

Author Response

Dear reviewer,

Thank you very much for the suggestions, which made the manuscript better.

The manuscript has been given to MDPI's English editor for revision to make the sentences more fluent and logical.

1.As you mentioned, the sample of the study is a limitation. It limited the representativeness of the findings. I have revised the content in the limitation section, adding discussions on convenience sample, self-report measures, etc.

2.In the introduction section, according to your suggestions, I have added some publications on the incidence of cyberbullying victimization in China, and analyzed the reasons that may cause the incidence to vary greatly. " The prevalence of cyberbullying victimization varied greatly among Chinese adolescents and children. A study of urban secondary school students in Guangdong province, China, reported a cyberbullying victimization rate of 44.5% [13], while the rate in rural primary and secondary school students in Shandong province, China, was 11.4% [14]. Variations in research tools and instruments, demographic characteristics of the survey sample, economic development, geographical location, cultural backgrounds, and Internet penetration rates may contribute to different cyberbullying victimization rates [15]. "

3.In the introduction section, I revised the following sentence "Empirical studies have showed that cyberbullying victimization is positively predicted depression among adolescents. " to" Meta-analyses and longitudinal studies have shown that cyberbullying victimization is a predictor of depressive symptoms among adolescents [16,27,28].”.

4.In the introduction section, as you mentioned, Line 95: “Bullying is a negative event that occurs in interpersonal relationships.” This study focuses on the relationship between cyberbullying victimization and depression, so I have revised this paragraph to "Cyberbullying is a negative event that occurs in interpersonal relationships via electronic means. "

5.In the participants and procedures section, in accordance with your suggestion, I provide further information on the recruitment of the sample and data collection. The additions are as follows: Using a convenience sampling method, we collected data from the primary school attached to Central China Normal University in Hubei province, which is one of Chinese top 100 primary schools. The students in this primary school are the children of Central China Normal University staff and residents of the nearby community. We reached out to the vice-principals of the primary school. As young children may have a limited cognitive ability to read and understand the questionnaire content, we covered all grade three to six students at this primary school. We recruited eight pre-trained psychology graduate students as experimenters. The training included learning the questionnaire content, responses to inquiries, interpersonal communication, and coding the questionnaire. Then, the experimenters collected the paper–pencil questionnaires at the primary school in April 2021. This study involving human participants was reviewed and approved by the Ethical Committee of Central China Normal University for Compliance with Ethical Standards (CCNU-IRB-201611025b).

6.As for the psychological support and intervention for children after asking them about cybervictimization. We have an agreement with the school's vice principal on possible harm to the child, we offer free group counseling to victims of cyberbullying and participants with high depressive symptoms. During the survey, we also had licensed psychologist (affiliated with the School of Psychology, Central China Normal University) to provide psychological support services. At the same time, we also provided participants with our AI service platform for psychological assistance, where children can have self-help counseling and can contact our psychological counselors on the platform.

  1. In the measures section, according to your suggestion, I have added references to the validation of the scales I used (i.e., cybervictimization; Self-perceived social competence).
  2. In the discussions section, according to your suggestion, I added some sentences to explain grade differences among cyberbullying victimization and the interaction of gender and grade in depression. The additions are as follows: Our study found that the rate of cyberbullying victimization among Chinese children was 31.1%. There was no significant gender difference in cyberbullying victimization of children. Cyberbullying victimization increased with grade, and the sixth graders experienced significantly more cyberbullying victimization than other children. As the grade increases, children spend more time online and use more information and communication technology. At the same time, with the development of technological innovation and Internet applications, children are more likely to be involved in cyberbullying and become victims of cyberbullying. Previous studies of cyberbullying victimization have shown inconsistent findings regarding gender differences, which may be related to variations in the study features (e.g., modality of cyberbullying, cyberbullying scale type, representativeness of sampling, region of sampling, reporting time frame) [70]. Our research found that boys in lower grades reported more serious depressive symptoms than girls, while girls in upper grades reported more serious depressive symptoms than boys. Boys’ social support networks are less intimate and supportive than girls' [71]. Girls showed more positive emotions than boys in middle childhood [72]. Boys were more likely to interact with information and communication technology, increasing their risk of cyberbullying victimization. These may explain why depressive symptoms are higher in lower-grade boys. Girls in upper grades reached puberty earlier than boys, which is an important factor affecting girls’ depression [73]. Moreover, girls in upper grades are more relationally oriented and exhibit greater affiliative needs [74]. As the grade increases, girls are more likely to use social media and become depressed due to cyberbullying victimization than boys.

Thank you again for your good advice to make this manuscript even better!

Round 2

Reviewer 1 Report

Comments and Suggestions for Authors

Thank you for your answer, everything is okay now.